# PREFER TO CLASSIFY: IMPROVING TEXT CLASSIFIER VIA PAIR-WISE PREFERENCE LEARNING

## ABSTRACT

The development of largely human-annotated benchmarks has driven the success of deep neural networks in various NLP tasks. These benchmarks are collected by aggregating decisions made by different annotators on the target task. Aggregating the annotated decisions via majority is still used as a common practice, despite its inevitable limitation from simple aggregation. In this paper, we establish a novel classification framework, based on task-specific *human preference* between a pair of samples, which provides an informative training signal to capture fine-grained and complementary task information through pair-wise comparison. Hence, it improves the existing instance-wise annotation system by enabling better task modeling from learning the relation between samples. Specifically, we propose a novel multi-task learning framework, called prefer-to-classify (P2C), to effectively learn human preferences in addition to the given classification task. We collect human preference signals in two ways: (1) extracting relative preferences *implicitly* from annotation records (for free) or (2) collecting subjective preferences *explicitly* from (paid) crowd workers. In various text classification tasks, we demonstrate that both extractive and subjective preferences are effective in improving the classifier with our preference learning framework. Interestingly, we found that subjective preference shows more significant improvements than extractive preference, revealing the effectiveness of explicit modeling of human preferences. Our code and preference dataset will be publicly available upon acceptance.

## 1 INTRODUCTION

The recent success of natural language processing (NLP) systems has been driven by, among other things, the construction of largely human-annotated benchmarks, like GLUE (Wang et al., 2019) or SQuAD (Rajpurkar et al., 2016). Nevertheless, current NLP benchmarks often include problems occurring in their construction process, such as annotation artifacts (Gururangan et al., 2018) or spurious patterns (Kaushik & Lipton, 2018). To alleviate these issues, various approaches have been recently proposed to construct more robust and effective benchmarks via human-in-the-loop with model (Kiela et al., 2021; Yuan et al., 2021; Liu et al., 2022) or adversarial sample mining (Kaushik et al., 2020; Nie et al., 2020; Potts et al., 2021). Despite such a careful selection of samples to annotate, it is relatively under-explored how to aggregate the annotations and assign the label, to fully exploit the advantage of these benchmarks.

For example, most NLP data collection still follows a long-standing custom for annotation voting called majority voting (Snow et al., 2008; Hovy et al., 2013) that aggregates multiple annotators' judgments into majority-voted ones. This labeling with majority voting, however, inevitably discards the valuable information embedded in multiple annotators' assessments and their disagreements, such as the inherent difficulty of instance (Pavlick & Kwiatkowski, 2019) or uncertainty from the task's subjectivity (Alm, 2011). As the modern NLP systems are extending the interest to a greater variety of social issues and subjective tasks (Uma, 2021), such as humor detection (Simpson et al., 2019) and racist language detection (Larimore et al., 2021), the capability of modeling the fine-grained, distributional opinions from multiple annotators becomes more important.

To address the limitation of the simple labeling method, various approaches have been recently proposed, such as label smoothing (Fornaciari et al., 2021; Leonardelli et al., 2021). However, they are still limited by the discretized annotation space and the limited number of annotators, resulting in

(a) Pair-wise preference signals    (b) Alignment to human annotations    (c) Improved text classification

Figure 1: (a) Example of a pair-wise preference in the sentiment classification. (b) Effect of preference learning. It makes the classifier capture the fine-grained task information; *e.g.*, predictions of classifier become more aligned with human annotations. Test samples are divided into Hard, Normal, Easy based on the annotators' disagreement. (c) Improvement from the collected preference and P2C in various aspects, *e.g.,* better accuracy and calibration. More results are presented in Section 4.2.

coarse-grained modeling of the task. Hence, it inspires us to investigate a new and complementary direction for capturing fine-grained task information, by relatively ordering a pair of two texts and better calibrating them with respect to the task, using *human preference*.

**Contribution.** In this paper, we establish a new classification framework based on *human preference* between pair of samples, *e.g.*, which text is more positive for sentiment classification (see Figure 1(a)), in addition to the task annotations. Learning with human preference has been demonstrated in multiple domains, including reinforcement learning (Christiano et al., 2017) and generative models (Ziegler et al., 2019), by training the model to follow human behavior and achieve the complex goal with better modeling of the task. While this direction is under-explored in the classification regime, it could provide an informative training signal by capturing the complementary task information through 'pair-wise' comparison that cannot be captured with 'instance-wise' evaluation (see Figure 1(b)). Hence, it would effectively improve the classifier, as shown in Figure 1(c).

Specifically, we propose a novel multi-task learning framework, coined prefer-to-classify (P2C), to effectively train the model from both classification and preference learning tasks. We introduce diverse multiple preference heads beside the classification head of the model to learn from preference labels. Then, we apply a consistency regularization between them for imposing the model to have higher confidence in classification with the preferred samples. We also develop two advanced sampling schemes to select more informative text pairs during the training.

To train P2C, we collect two types of human preference labels: *extractive preference* and *subjective preference*. Extractive preference is constructed from the existing annotation records in datasets 'without additional cost'; if one sample has been less voted than the other, we treat the latter as a relatively higher preference between the two samples. One may argue that the extracted preferences are somewhat artificial (yet for free) and implicit signals, as they are not obtained from direct comparison by human. To alleviate this, we also collect subjective preferences for 5,000 pairs of texts from (paid) crowd workers by directly asking them which text is more preferred to the task label.

We demonstrate the effectiveness of preference learning via P2C in addition to given task-specific labels, on both extractive and subjective preference labels. In six text classification datasets, P2C with extractive preference exhibited 7.59% and 4.27% relative test error reduction on average, compared to the training with majority voting and the previous best method to learn using annotation records, respectively. Moreover, our newly-collected subjective preference labels show clear advantages over the extractive ones, not only with the improvement in task performance but also with better calibration and task modeling; for example, 6.09% of expected calibration error while 9.19% from the same number of task labels. Overall, our work highlights the effectiveness of pairwise human preference for better task learning; we suggest that NLP benchmarks should include annotation records, instead of just providing majority-voted labels, or collect human preferences.

## 2   IMPROVING TEXT CLASSIFICATION VIA PREFERENCE LEARNING

In this section, we present prefer-to-classify (P2C), a new preference learning framework for text classification. Our main idea is to take advantage of the preference between two input samples to

train the classifier by providing the complementary learning signal from a pair-wise comparison that cannot be captured from the instance-wise task labels. In Section 2.1, we describe the problem settings and preliminaries. Then, in Section 2.2, we elaborate on the specific components of P2C for learning from human preferences. Figure 2 presents the overview of our framework.

## 2.1 PRELIMINARIES

**Problem description.** We describe the problem setup of our interest under a text classification scenario with $K$ classes. Let $\mathcal{D}$ denote the given training dataset consisting of tuples $(\mathbf{x}, y_{\texttt{task}}) \in \mathcal{D}$ where $\mathbf{x} = [x_1, \ldots, x_L]$ is the sequence of input tokens $x_i$, and $y_{\texttt{task}}$ is the target task label. Our goal is to train a classifier $f_\theta := W_{\texttt{task}} \circ g_\phi$, composed with Transformer-based language model backbone $g_\phi$ (*e.g.*, BERT (Devlin et al., 2019)) and a random initialized classification head $W_{\texttt{task}}$, to minimize the task-specific loss $\mathcal{L}_{\texttt{task}}$ such as a cross-entropy loss where $p(\mathbf{x}) = \text{Softmax}\big(f_\theta(\mathbf{x})\big)$. The common standard to determine the target label is one-hot labeling to the majority voted one *i.e.*, $y_{\texttt{task}} \in [0, 1]^K$, which aggregates multiple annotations on one single sample. To address the limitation of the majority voting, different labeling approaches have recently been explored by utilizing annotators' disagreements for the training, such as soft-labeling (Fornaciari et al., 2021).

**Preference learning.** In this paper, we use a human preference between two data instances as a complementary learning signal to train the classifier. Specifically, the preference signals reflect the relative suitability between the two input samples concerning the given task. We assume that the human preferences, *i.e.*, preference labels, of the given dataset are available.[1] Then, our goal is training a preference predictor to learn from the given human preferences, by predicting which one among the two input samples is more preferred. To this end, we formulate a preference learning as a supervised learning problem following the approaches in other domains such as reinforcement learning and generative modeling (Christiano et al., 2017; Ziegler et al., 2019; Lee et al., 2021). Given a pair of two different input tokens $(\mathbf{x}^0, \mathbf{x}^1)$ and task label $y_{\texttt{task}}$, a preference label $y_{\texttt{pref}}$ is additionally given; it indicates which input is preferred considering $y_{\texttt{task}}$, *i.e.*, $y_{\texttt{pref}} \in \{0, 1, 0.5\}$, where 1 indicates $\mathbf{x}^1 \succ \mathbf{x}^0$ (*i.e.*, $\mathbf{x}^1$ is preferred than $\mathbf{x}^0$), 0 indicates $\mathbf{x}^0 \succ \mathbf{x}^1$, and 0.5 implies an equally preferable case. Each preference label is stored in a dataset $\mathcal{D}$ as a quadruplet $(\mathbf{x}^0, \mathbf{x}^1, y_{\texttt{task}}, y_{\texttt{pref}})$. Then, we predict a preference using the preference predictor $h_\psi$ following (Bradley & Terry, 1952):

$$P_\psi[\mathbf{x}^1 \succ \mathbf{x}^0; y_{\texttt{task}}] = \frac{\exp\big(h_\psi(\mathbf{x}^1, y_{\texttt{task}})\big)}{\sum_{i \in \{0,1\}} \exp\big(h_\psi(\mathbf{x}^i, y_{\texttt{task}})\big)} \tag{1}$$

where $\mathbf{x}^i \succ \mathbf{x}^j$ implies that input $i$ is preferable to input $j$. The underlying assumption of this model is that the probability of preferring an input depends exponentially on its output. Then, the preference predictor $h_\psi$ is trained through supervised learning with the given human preferences, by minimizing the binary cross-entropy loss as follow[2]:

$$\mathcal{L}_{\texttt{pref}} = - \underset{\substack{(\mathbf{x}^0, \mathbf{x}^1, y_{\texttt{task}}, y_{\texttt{pref}}) \\ \sim \mathcal{D}}}{\mathbb{E}} \Big[ y_{\texttt{pref}} \log P_\psi[\mathbf{x}^1 \succ \mathbf{x}^0; y_{\texttt{task}}] + (1 - y_{\texttt{pref}}) \log P_\psi[\mathbf{x}^0 \succ \mathbf{x}^1; y_{\texttt{task}}] \Big] \tag{2}$$

## 2.2 PREFER-TO-CLASSIFY (P2C)

Next, we present our specific techniques to train the classifier with preference labels: (*a*) multi-task learning of classification and preference learning, (*b*) consistency regularization between classification and preference learning, and (*c*) informative pair sampling using disagreement or inconsistency.

**Multi-task learning with preference labels.** To effectively learn from the given preference label $y_{\texttt{pref}}$ and the task label $y_{\texttt{task}}$, we train the classifier $f_\theta$ via a multi-task learning (Ruder, 2017; Sener & Koltun, 2018) of both classification and preference learning. Specifically, we model the preference predictor $h_\psi$ in Eq. 1 upon the classifier $f_\theta$ similar to the case of $W_{\texttt{task}}$. The preference prediction head $W_{\texttt{pref}}$ is added on the output of Transformer backbone $g_\phi(\mathbf{x})$ and task label $y_{\texttt{task}}$, *i.e.*, $h_\psi(\mathbf{x}, y_{\texttt{task}}) = W_{\texttt{pref}} \circ [g_\phi(\mathbf{x}); y_{\texttt{task}}]^3$ where $f_\theta(\mathbf{x}) = W_{\texttt{task}} \circ g_\phi(\mathbf{x})$.

---

[1]See Section 3 for two different practical scenarios on how preference labels are collected.

[2]Equally preferable case is learned by imposing the same coefficients ($y_{\texttt{pref}} = 1 - y_{\texttt{pref}} = 0.5$) in Eq. 2.

[3]$[g_\phi(\mathbf{x}); y_{\texttt{task}}]$ means the concatenation between $g_\phi(\mathbf{x})$ and $y_{\texttt{task}}$.

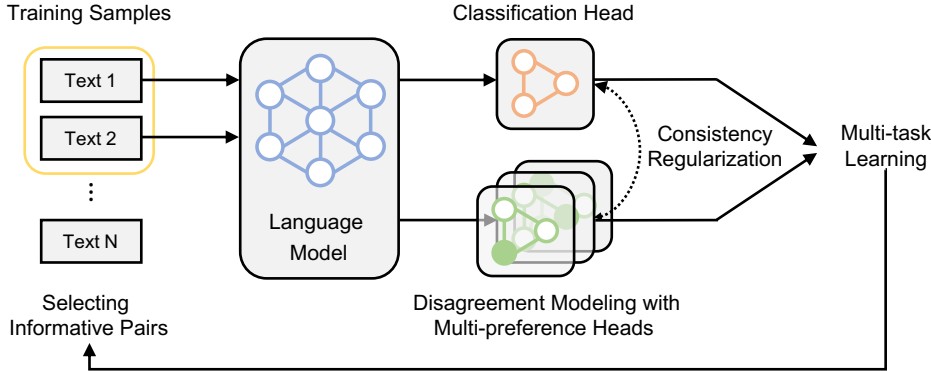

Figure 2: Illustration of **P**refer-**to**-**C**lassify (P2C) framework.

**Preference modeling with diverse multi-preference heads.** In addition, we introduce multiple preference heads $\{W_{\text{pref}}^{(t)}\}_{t=1}^T$ to fully exploit the given preference labels, through the advantage of classifier ensemble (Ganaie et al., 2021). Each preference head is independently random initialized and trained with $\mathcal{L}_{\text{pref}}$ in Eq. 2. Since the different initialization is limited to impose the diversity between $\{W_{\text{pref}}^{(t)}\}_{t=1}^T$ during the training, we add a regularization $\mathcal{L}_{\text{div}}$ to encourage the diverse prediction for each preference head by maximizing KL-divergence between them (Wang et al., 2021):

$$\mathcal{L}_{\text{div}} = \frac{-1}{T-1} \sum_{j=1, j \neq i}^T D_{\text{KL}}\big(P_{\psi^{(i)}}(\mathbf{x}^1, \mathbf{x}^0; y_{\text{task}}) || P_{\psi^{(j)}}(\mathbf{x}^1, \mathbf{x}^0; y_{\text{task}})\big) \tag{3}$$

where $P_\psi(\mathbf{x}^1, \mathbf{x}^0; y_{\text{task}})$ is the predictive distribution of the preference predictor $h_\psi$, *i.e.*, $P_\psi(\mathbf{x}^1, \mathbf{x}^0; y_{\text{task}}) = [P_\psi[\mathbf{x}^1 \succ \mathbf{x}^0; y_{\text{task}}], P_\psi[\mathbf{x}^0 \succ \mathbf{x}^1; y_{\text{task}}]]$. Overall, we train the classifier with the following multi-task learning objective $\mathcal{L}_{\text{multi}}$ under hyper-parameter $\lambda_{\text{div}}$:

$$\mathcal{L}_{\text{multi}} = \mathcal{L}_{\text{task}} + \mathcal{L}_{\text{pref}}^{\text{all}} + \lambda_{\text{div}}\mathcal{L}_{\text{div}} \tag{4}$$

where $\mathcal{L}_{\text{pref}}^{\psi^{(t)}}$ indicate the preference learning objective with each head $\psi^{(t)}$ and $\mathcal{L}_{\text{pref}}^{\text{all}} = \sum_{t=1}^T \mathcal{L}_{\text{pref}}^{\psi^{(t)}}$.

**Consistency regularization between classification and preference learning.** Even though multi-task learning is an effective way to train the model, it is still unclear whether or not the model can capture the relations between the two tasks explicitly. Accordingly, we hypothesize that *a more preferred instance should have a higher confidence from the classifier*, *i.e.*, $p_y(\mathbf{x}^1) > p_y(\mathbf{x}^0)$ if $\mathbf{x}^1 \succ \mathbf{x}^0$ with the given task label $y$. Hence, to impose the model explicitly follows this intuition, we further propose a consistency regularization between the two tasks as follow:

$$\mathcal{L}_{\text{cons}} = y_{\text{pref}} \max\{0, p_y(\mathbf{x}^1) - p_y(\mathbf{x}^0)\} + (1 - y_{\text{pref}}) \max\{0, p_y(\mathbf{x}^0) - p_y(\mathbf{x}^1)\} \tag{5}$$

Additionally, when the degree of preference is explicitly provided, *i.e.*, $y_{\text{pref}} \in [0, 1]$ (see Section 4.1 of extractive preference case) rather than $y_{\text{pref}} \in \{0, 1, 0.5\}$, we further extend this consistency regularization with margin $m$ which represents the degree of preference:

$$\mathcal{L}_{\text{cons}} = y_{\text{pref}} \max\{0, m - \Delta p_y(\mathbf{x}^1, \mathbf{x}^0)\} + (1 - y_{\text{pref}}) \max\{0, \Delta p_y(\mathbf{x}^1, \mathbf{x}^0) - m\} \tag{6}$$

where $\Delta p_y(\mathbf{x}^1, \mathbf{x}^0) = p_y(\mathbf{x}^0) - p_y(\mathbf{x}^1)$. We note that the previous consistency regularization Eq. 5 becomes the special case of Eq. 6 with $m = 0$. Overall, our training loss of the classifier is as follow:

$$\mathcal{L}_{\text{train}} = \mathcal{L}_{\text{multi}} + \lambda_{\text{cons}}\mathcal{L}_{\text{cons}} \tag{7}$$

where $\lambda_{\text{cons}}$ is a hyper-parameter.

**Selecting informative pairs.** As the number of pairs of samples $(\mathbf{x}^0, \mathbf{x}^1)$ is proportional to the square of the number of training samples, it is difficult to obtain the preference label for all possible pairs, and even harder to learn from them even if we have all the preference labels. Hence, we propose the following advanced sampling scheme to maximize preference learning's effectiveness during training: (1) *Disagreement-based* sampling, which selects pairs of instances with high variance across multiple preference predictors $\{h_{\psi^{(i)}}\}_{i=1}^T$, and (2) *Inconsistency-based* sampling, which seeks to reduce the mismatched pairs with high consistency loss $\mathcal{L}_{\text{cons}}$ in Eq. 5. We evaluate the effects of these sampling methods in Appendix C.

Table 1: Examples of the collected extractive and subjective preference sets.

| | |
|---|---|
| **A**: I got 3 veggies and a side of fries for over a 11 dollars if you like homecooked food | **B**: She listened to my ideas, asked questions to get a better idea about my style, and was excellent at offering advice as if I were a total pleb. |
| Sentiment: Positive, Extractive Preference: **A ≻ B**, Subjective Preference: **B ≻ A** | |
| **A**: This was the best movie. Movie was entertaining but not as good as the original | **B**: The restaurant was not busy, but everything was ready for a big crowd if needed. We love this place. |
| Sentiment: Positive, Extractive Preference: **No preference**, Subjective Preference: **B ≻ A** | |
| **A**: Drove 15 miles out of my way for them to tell me on the speaker at 9:58 it's closed. i am pissed | **B**: Wow, the service was like I was staying at San Quentin for 2 to 4 years. |
| Sentiment: Negative, Extractive Preference: **B ≻ A**, Subjective Preference: **A ≻ B** | |
| **A**: I never get it for the holiday. | **B**: The casino was full, with the slots hitting. |
| Sentiment: Neutral, Extractive Preference: **A ≻ B**, Subjective Preference: **B ≻ A** | |

## 3 COLLECTION OF PREFERENCE LABELS

In this section, we provide two different ways of collecting human preferences to prepare the ground-truth preference labels for our training: *extractive* and *subjective* preferences.

### 3.1 EXTRACTING IMPLICIT PREFERENCES FROM DATA ANNOTATION RECORDS

The first method is to extract human preference signals from the existing datasets without extra cost. Our high-level assumption is that annotation records of each data, which are naturally gathered during the construction of dataset, implicitly encode the preference between data samples. For example, if one sample has higher voting (9 out of 10), *i.e.*, less disagreement, than the other sample (6 out of 10), we assume that the former has a relatively higher preference. We call this implicit human preference as *extractive preference*. Since the extractive preference is derived from existing sources of data, it can be obtained for any pair of samples without additional cost.

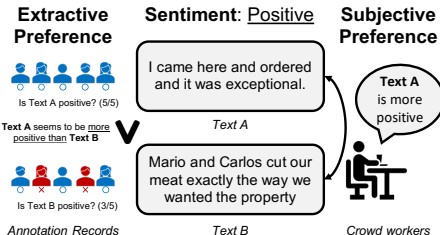

Figure 3: Comparison between extractive and subjective preference labels.

### 3.2 COLLECTING SUBJECTIVE PREFERENCES FROM CROWD WORKERS

Although the extractive preferences are easy to obtain and intuitive, they are sub-optimal since they are not obtained from the direct comparison of the pair of sentences by human labeler. Hence, to the best of our knowledge, we collect the first preference dataset for the text classification. Our dataset is collected based on paired samples from DynaSent-R2 dataset (Potts et al., 2021) for sentiment classification task. To be specific, we gather the subjective preference of the pairs by asking crowd workers to answer "*which sentence is more positive (neutral, or negative)?*" using Amazon's Mechanical Turk crowd-sourcing platform (Crowston, 2012). Then, each worker should select one of the two sentences or answer "No Preference". Following (Nie et al., 2020), we initially provide each pair of sentences to two crowd workers. If two workers give the same preference label, this pair is labeled with that. If they disagree, we ask a third crowd worker to break the tie. If they still fail to reach a consensus, this pair is labeled with "No Preference".

Under this procedure, we first gather 1,000 subjective preference labels of randomly selected pairs of sentences. Then, we dynamically collect the additional subjective preference labels to maximize the information of collected pairs, motivated by the recent dynamic benchmark constructions (Kiela et al., 2021; Nie et al., 2020). Namely, we first train the model with existing subjective preference labels. Then, we find the most informative pairs in the aspect of trained model, using the disagreement-based sampling introduced in Section 2.2 and query their preference labels in next stage. We select an equal number of pairs for each class to balance the label distribution. Overall, starting with 1,000 random pairs, we collect the preference of 2,000 pairs at each round and iterate this procedure for 2 rounds, *i.e.*, a total of 5,000 pairs' subjective preference labels are collected. The used interface for gathering the preference label and more detailed statistics are presented in Appendix B. Also, Table 1 shows

some examples in our extractive and preference sets. Here, the advantage of subjective preference compared to extractive one could be observed by providing more accurate pair-wise preference.

## 4 EXPERIMENTS

In this section, we validate the effectiveness of the proposed P2C with the extractive preference labels (for free) in Section 4.1 and the subjective preference labels (payed) in Section 4.2, respectively.

### 4.1 EXPERIMENTS WITH EXTRACTIVE PREFERENCE FROM ANNOTATION RECORDS

**Datasets.** To extract preferences from the annotation records, we extensively investigate the publicly available datasets providing such information and use the following six text classification datasets. All datasets have the annotation records from 5 annotators for each sample. DynaSent (Potts et al., 2021) is a sentiment classification benchmark with ternary (positive/negative/neutral) sentiments. It is dynamically constructed through multiple iterations of training a classifier model and finding its adversarial text inputs. We use the dataset from the first round, (1) *DynaSent-R1*, and from the second round, (2) *DynaSent-R2*, the dataset for our experiments. Standford politeness corpus (Danescu-Niculescu-Mizil et al., 2013) is a binary classification benchmark for predicting whether the given sentence is polite or impolite. Since there are two different input domains within this benchmark, we split them into two different datasets: (3) *Polite-Wiki* from Wikipedia, and (4) *Polite-SE* from StackExchange, following the original paper. Offensive agreement dataset (Leonardelli et al., 2021) is a binary classification benchmark for predicting whether the given sentence is offensive or not. Since some of the original samples are not available anymore publicly, we only utilize the available samples while keeping the setups of original dataset: (5) *Offensive*. (6) *MultiNLI* (Williams et al., 2018) is a crowd-sourced collection of sentence pairs annotated with textual entailment information. As the only validation set includes the annotation records, we split it into 8:1:1 for training, validation, and test sets. Detailed description and pre-processing procedures are presented in Appendix A.1.

**Baseline methods.** We first compare the proposed P2C to a naïve training using majority voting without consideration of disagreement, denoted by (a) *Vanilla*. Then, since our method with extractive preference can be viewed as a new way to utilize the annotators' disagreement from the annotation records, we compare P2C with a wide range of disagreement learning methods, as listed as follows; (b) *Soft-labeling* (Fornaciari et al., 2021): using the probabilistic distribution of annotations as soft labels for training; (c) *Margin* (Sharmanska et al., 2016): training the model with hinge loss by setting a margin proportional to the annotators' agreements; (d) *Filtering* (Leonardelli et al., 2021): removing the training samples with a high disagreement. Following (Leonardelli et al., 2021), we discard the samples with 3 agreements among 5 annotators, and use majority voting for the others; (e) *Weighting* (Uma et al., 2021): using weighted cross-entropy with smaller weights for the samples with high disagreements; (f) *Multi-annotator* (Davani et al., 2022): training the multiple classification heads for each annotation and using its ensemble for the evaluation. Furthermore, since we train the model with pair of samples, we also consider the baseline considering a pair-wise training, (g) Class-wise Self-Knowledge Distillation (*CS-KD*) (Yun et al., 2020): adding regularization between sample pair that forces the similar predictive distribution between the same class samples. More details about baselines are described in Appendix A.2.

**Training details.** All the experiments are conducted by fine-tuning RoBERTa-base (Liu et al., 2019) using Adam optimizer (Kingma & Ba, 2015) with a fixed learning rate 1e-5 and the default hyper-parameters of Adam. For all text classification tasks, the model is fine-tuned using the specified disagreement learning method with batch size 16 for 20 epochs. In the case of P2C, we use $T = 3$ preference heads $\{W_{\texttt{pref}}^{(i)}\}_{i=1}^{T}$ and 2-layer MLPs for each $W_{\texttt{pref}}$. We choose hyper-parameters from a fixed set of candidates based on the validation set: $\lambda_{\texttt{cons}}, \lambda_{\texttt{div}} \in \{1.0, 0.1\}$. We sample the pair of instances with the same majority voted labels for the efficiency, and apply the consistency loss with margin (Eq. 6) by using the difference of annotation as the margin $m$. More details and experimental supports for the design choices can be found in Appendix A.3 and C, respectively.

**Results.** We compare P2C with various disagreement learning schemes to fine-tune RoBERTa-base classifier for each dataset. Table 2 summarizes the results on six text classification datasets. Remarkably, P2C consistently outperforms the baseline methods for all six datasets. To be specific, P2C exhibits 7.59% relative test error reduction compared to the vanilla method in the average.

Table 2: Test accuracy of finetuned RoBERTa classifiers with each annotation method on 5 different text classification datasets. All the values and error bars are mean and standard deviation across 5 random seeds. The best and the second best results are indicated in **bold** and underline, respectively.

| Method | Offensive | Polite-Wiki | Polite-SE | MNLI | DynaSent-R1 | DynaSent-R2 |
|---|---|---|---|---|---|---|
| Vanilla | $75.88_{\pm 0.72}$ | $89.35_{\pm 1.53}$ | $70.00_{\pm 1.49}$ | $81.92_{\pm 0.70}$ | $80.43_{\pm 0.30}$ | $71.23_{\pm 1.05}$ |
| Soft-labeling | $76.08_{\pm 1.44}$ | $89.57_{\pm 1.76}$ | $70.35_{\pm 1.68}$ | $\underline{82.67}_{\pm 0.50}$ | $81.10_{\pm 0.33}$ | $\underline{72.15}_{\pm 1.59}$ |
| Margin Loss | $\underline{76.67}_{\pm 1.18}$ | $88.51_{\pm 0.93}$ | $\underline{70.51}_{\pm 1.16}$ | $81.41_{\pm 0.63}$ | $80.42_{\pm 0.23}$ | $69.27_{\pm 0.98}$ |
| Filtering | $76.13_{\pm 1.18}$ | $89.50_{\pm 0.87}$ | $68.28_{\pm 2.43}$ | $82.13_{\pm 0.67}$ | $80.38_{\pm 0.34}$ | $69.86_{\pm 0.78}$ |
| Weighting | $76.17_{\pm 1.18}$ | $89.65_{\pm 1.46}$ | $68.38_{\pm 1.67}$ | $82.48_{\pm 0.49}$ | $80.21_{\pm 0.41}$ | $71.81_{\pm 1.12}$ |
| Multi-annotator | $76.50_{\pm 1.98}$ | $\underline{89.88}_{\pm 1.82}$ | $69.39_{\pm 2.84}$ | $82.61_{\pm 0.70}$ | $\underline{81.14}_{\pm 0.55}$ | $71.97_{\pm 1.25}$ |
| CS-KD | $75.75_{\pm 0.66}$ | $89.65_{\pm 1.84}$ | $70.10_{\pm 1.29}$ | $82.32_{\pm 0.23}$ | $80.63_{\pm 0.27}$ | $71.81_{\pm 0.67}$ |
| P2C (Ours) | $\mathbf{77.81}_{\pm 0.21}$ | $\mathbf{91.06}_{\pm 0.64}$ | $\mathbf{71.21}_{\pm 0.93}$ | $\mathbf{83.15}_{\pm 0.29}$ | $\mathbf{81.27}_{\pm 0.46}$ | $\mathbf{73.06}_{\pm 0.31}$ |

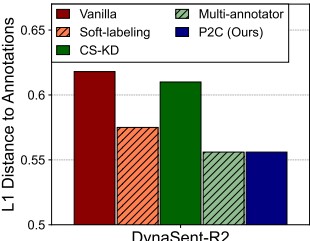

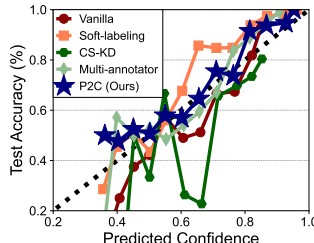

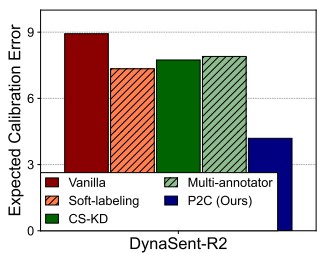

(a) Disagreement Prediction (↓)   (b) Reliability Diagram (↗)   (c) Expected Calibration Error (↓)

Figure 4: Additional experiments with P2C on DynaSent-R2. (a) Average L1 distance between the predictions and the soft labels obtained from the annotation records. The lower distance (↓) means better alignment with annotators. (b) Reliability diagram shows accuracy as a function of confidence. Perfect calibration is plotted by dashed diagonals (↗). (c) Expected Calibration Error (ECE) to quantitatively measure the calibration of the classifier. The lower ECE (↓) means better calibration.

Furthermore, compared to the previous best disagreement learning method of each dataset, P2C exhibits 4.27% relative test error reduction on average. These results show that extractive preferences successfully provide complementary training signals to the classifier from the pair-wise preference, and demonstrate the effectiveness of P2C as a training method to utilize the annotation records.

Next, on DynaSent-R2, we conduct additional experiments to verify depthly how P2C improves the classifier. We first check whether the prediction of the trained model with P2C is similar to the annotators' judgment. Specifically, we compare the L1 distance between the predictions and the soft labels obtained from the data annotation records in Figure 4(a). We verify that P2C achieves the lowest distance to the soft labels, showing the validity of our preference learning for better modeling of given task. Moreover, we verify that a calibration of the classifier is also improved as a result of pair-wise preference modeling. To be specific, we first provide a reliability diagram (Yun et al., 2020), which plot the expected sample accuracy as a function of confidence of classifier in Figure 4(b). We remark that the plotted identity function (dashed diagonal) implies perfect calibration (Guo et al., 2017), and our method is the closest one among the baselines. This calibration effect of P2C is also verified through the additional quantitative metric, Expected Calibration Error (ECE), in Figure 4(c). Here, we commonly adopt the temperature scaling to measure ECE following (Guo et al., 2017).

To verify the effectiveness of each component of P2C, we perform the ablation experiments and the results are presented in Table 3. It is observable that diverse multi-preference heads improve the effectiveness of preference labels with better modeling compared to the single preference head (*2-4th rows*). In addition, consistency regularization between classification and preference heads enables the classifier to fully utilize the pair-wise training signal to solve the task, hence the performance is significantly improved (*5th row*). The performance is further improved by selecting the informative pairs during the training (*6th row*). More results are presented in Appendix C.

### 4.2 EXPERIMENTS WITH SUBJECTIVE PREFERENCE FROM CROWD-WORKERS

**Setups.** As described in Section 3.2, we collect subjective preference labels of the samples on DynaSent-R2, and use them to fine-tune RoBERTa-base. All other training setups and details are

Table 3: Ablation study with each component of P2C. Test accuracy of finetuned RoBERTa classifiers on DynaSent-R2 and Offensive are compared. All the values and error bars are mean and standard deviation across 5 random seeds. The best results are indicated in **bold**.

| Method | $T$ | $\mathcal{L}_{\texttt{task}}$ | $\mathcal{L}_{\texttt{pref}}$ | $\mathcal{L}_{\texttt{div}}$ | $\mathcal{L}_{\texttt{cons}}$ | Sampling | DynaSent-R2 | Offensive |
|---|---|---|---|---|---|---|---|---|
| Vanilla | - | ✓ | - | - | - | - | $71.23_{\pm1.05}$ | $75.88_{\pm0.72}$ |
| Preference | 1 | ✓ | ✓ | - | - | - | $71.84_{\pm0.78}$ | $75.90_{\pm1.15}$ |
| | 3 | ✓ | ✓ | - | - | - | $71.92_{\pm0.66}$ | $76.43_{\pm0.32}$ |
| | 3 | ✓ | ✓ | ✓ | - | - | $72.05_{\pm1.30}$ | $76.67_{\pm1.38}$ |
| | 3 | ✓ | ✓ | ✓ | ✓ | - | $72.67_{\pm0.89}$ | $77.67_{\pm0.99}$ |
| P2C (Ours) | 3 | ✓ | ✓ | ✓ | ✓ | ✓ | $\mathbf{73.06}_{\pm0.31}$ | $\mathbf{77.81}_{\pm0.21}$ |

Table 4: Results of finetuned RoBERTa classifiers with different ways to obtain the labels on DynaSent-R2. $N_{\texttt{task}}$ and $N_{\texttt{pref}}$ indicate the number of used task labels and preference labels, respectively. $d_{\texttt{hard}}$ and $d_{\texttt{easy}}$ are the $l_1$ distance to annotations with hard and easy samples. Here, the difficulty is defined on the disagreement between annotators. All the values and error bars are mean and standard deviation across 5 random seeds. The best results are indicated in **bold**.

| Method | $N_{\texttt{task}}$ | $N_{\texttt{pref}}$ | $\text{Acc}_{\texttt{avg}}(\uparrow)$ | $\text{Acc}_{\texttt{hard}}$ / $\text{Acc}_{\texttt{easy}}(\uparrow)$ | $\text{ECE}(\downarrow)$ | $d_{\texttt{hard}}$ / $d_{\texttt{easy}}(\downarrow)$ |
|---|---|---|---|---|---|---|
| Vanilla | 7.5k | - | $69.03_{\pm1.29}$ | $59.33_{\pm2.57}$ / $80.00_{\pm1.22}$ | $9.25_{\pm1.39}$ | $0.856_{\pm0.01}$ / $0.405_{\pm0.03}$ |
| Task Labels | 12.5k | - | $71.17_{\pm1.35}$ | $57.86_{\pm2.31}$ / $\mathbf{84.21}_{\pm1.05}$ | $9.19_{\pm1.36}$ | $0.878_{\pm0.04}$ / $\mathbf{0.327}_{\pm0.02}$ |
| Extractive Preference | 7.5k | 5k | $71.36_{\pm1.19}$ | $61.16_{\pm1.91}$ / $83.11_{\pm1.78}$ | $6.75_{\pm0.78}$ | $0.847_{\pm0.03}$ / $0.351_{\pm0.03}$ |
| Subjective Preference | 7.5k | 5k | $\mathbf{71.74}_{\pm1.04}$ | $\mathbf{62.08}_{\pm0.94}$ / $83.01_{\pm1.27}$ | $\mathbf{6.09}_{\pm0.31}$ | $\mathbf{0.828}_{\pm0.02}$ / $0.356_{\pm0.02}$ |

same as described in Section 4.1 except that the consistency regularization (Eq. 5) is now used, since the explicit degree of preference is not available from the subjective preference. In this section, we mainly compare P2C with different types of the preference labels: *Extractive* and *Subjective* preference labels. But the subjective preference labels have been collected for limited number of pairs of sentences (5,000) due to the cost, they are not available for all pairs of samples unlike extractive one. Hence, we also limits the availability of extractive preference labels only for the same pairs of texts with subjective preference, to directly compare the effect of different preference labels.

**Results.** We first verify the effectiveness of collected subjective preference labels compared to other types of labels. To this end, we consider the scenario that the specific types of labels are additionally obtained on top of the existing task labels; task labels could be more collected with additional training samples or preference labels between the existing samples could be obtained.[4] Table 4 summarizes the experimental results. Here, it is observed that subjective preference labels is the most effective for improving the test accuracy ($\text{Acc}_{\texttt{avg}}$) along with better calibration. Remarkably, it is noticeable that the preference labels significantly improves the accuracy on relatively hard samples ($\text{Acc}_{\texttt{hard}}$) while the additional task labels are effective for the relatively easy samples.[5]

Furthermore, to clearly demonstrate the advantage of subjective preference labels compared to the extractive one, we conduct additional experiments in Figure 5; as subjective preferences are collected with multiple rounds ($N_{\texttt{pref}} = 1,000 \rightarrow 3,000 \rightarrow 5,000$), we evaluate the effectiveness of preference labels for all rounds in Figure 5(b), by controlling the number of preference labels of extractive preference and subjective preference. Remarkably, P2C with subjective preference outperforms P2C with extractive preference for all cases with only a few thousand samples. Also, the performance of P2C is further improved when we extend the number of available preference labels via pseudo-labeling as denoted in the dotted lines (See details in Appendix D). The advantage of subjective preference is clearly shown when we take a closer look at the evaluation results on more challenging scenario: the accuracy on the mis-predicted test samples by vanilla model, and alignments with annotations on both hard and easy samples. These results indicate that obtaining the explicit preference labels directly collected from annotators are better than extracting the implicit ones from the data annotation records.

---

[4]Here, to facilitate the experimental comparison, we assume that the annotation cost for preference labels stays the same as that for task labels, although it is arguable.

[5]We define the difficulty based on the disagreement of annotators, *i.e.,* more disagree indicates more difficult.

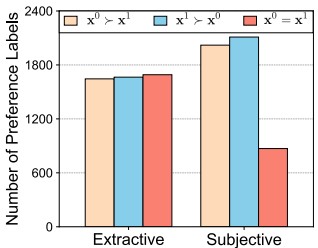 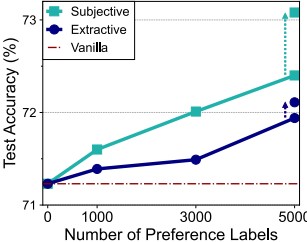 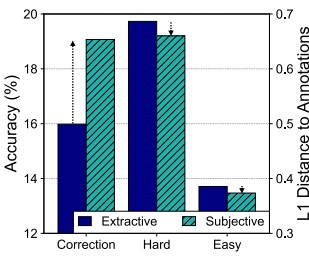

(a) Distribution of preference labels (b) Test accuracy improvements (c) Comparison of predictions

Figure 5: Comparison between two different types of preference labels (*Extractive* and *Subjective*) on DynaSent-R2. (a) Distribution of collected labels. $\mathbf{x}^0 \succ \mathbf{x}^1$ indicates that sample $\mathbf{x}^0$ is preferred than $\mathbf{x}^1$ when considering $y_{\texttt{task}}$. (b) Overall test accuracy increases as more preference labels are used upon the existing task labels. (c) Subjective preference shows better performance on the mis-predicted test samples by *Vanilla* and closer alignment to annotators' agreements.

## 5  RELATED WORKS

**Preference learning from human feedback.** Preference learning is about modeling the preference using a set of instances, associated with an order relation (Fürnkranz & Hüllermeier, 2010). Since it is much easier for humans to make relative judgments, i.e., comparing behaviors as better or worse, preference-based learning becomes an attractive alternative; hence, the extensive research has been conducted to address this problem by proposing different techniques to learn from human judgements (Bıyık et al., 2020; Chu & Ghahramani, 2005). One of the most representative fields that adopt preference-based learning is Reinforcement Learning (RL), to learn RL algorithms from the preferences rather than the explicit design of reward function (Wirth et al., 2017). After the successful scale-up of preference-based learning with deep neural networks (Christiano et al., 2017; Lee et al., 2021), this research direction has been extensively explored in other domains such as NLP (Stiennon et al., 2020; Ziegler et al., 2019) and computer vision (Kazemi et al., 2020), especially focused on the generation tasks, *e.g.,* text summarization and image generation. However, preference-based learning is yet under-explored for classification tasks, despite its great potential to provide the complementary training signals via pair-wise comparison of samples.

**Labeling beyond majority voting.** With the rapid advance of DNNs thanks to improvements in computational resources (Jouppi et al., 2017) as well as algorithmic breakthroughs (Devlin et al., 2019), the existing NLP benchmarks easily become obsolete and hence suffer to keep up with the model's development. To alleviate this, various approaches have been recently explored to construct more challenging and robust benchmarks using a human-in-the-loop (Nie et al., 2020) or dynamic benchmarking system (Kiela et al., 2021). But, it is relatively under-explored *how to annotate* for utilizing them maximally. Even most of the recent NLP benchmarks still follow a long-standing custom for their annotation; obtaining multiple annotators' judgements on the same data instances and aggregating them with majority voting (Hovy et al., 2013). This aggregation, however, has a risk of sacrificing the valuable nuances embedded in the annotators' assessments and their disagreements. Hence, various approaches have been recently explored to exploit this information better and successfully improve the performance of DNNs in various NLP tasks; for example, Fornaciari et al. (2021) constructs soft labels from annotation records. Also, Leonardelli et al. (2021) suggests to remove the samples with high disagreement. However, these methods are still limited due to the discretization of annotation space and the limited number of annotators. Hence, it inspires us to investigate an independent and complementary direction.

## 6  CONCLUSION

In this paper, we introduce task-specific human preferences between pairs of samples as a new and complementary data annotation to improve the existing text classification system, which relies on instance-wise annotations. To this end, we propose a novel multi-task learning framework, called prefer-to-classify (P2C), to effectively train the classifier from both task and preference labels, and demonstrate this framework under two different cases of human preference labels. We hope that our work could motivate other researchers for better data annotation and data usage in the future, *e.g.,* suggesting to include annotation records instead of just providing majority-voted labels, in order to better learn the task from disagreed annotations.

## ETHICS STATEMENT

While the advantage of the proposed method has been demonstrated, some limitations exist. For example, as we have assume the availability of task labels before applying our method, one could wonder that the relative importance between task labels and preference labels for constructing the datasets from scratch, *i.e.,* the order of priority for collecting annotations among task labels and preference labels. If the number of given task labels is too small, then the task labels can be more effective than subjective preference labels. However, we highlight that our P2C could still benefit this case with the extractive preference as shown in Table 12. Also, the gain from subjective preference labels is enlarged after enough task labels are collected, demonstrating the complementary effect of preference labels upon the task labels as we are motivated initially. Lastly, this new framework is expected to be more effective in the domains that input samples are hard to collect as it provides the additional task information, *e.g.*, medical domain.

## REPRODUCIBILITY STATEMENT

We describe the implementation details of the method in Section 4 and Appendix A.3. Also, we provide the details of the datasets and baselines in Appendix A.1 and A.2, respectively. We also provide our code in the supplementary material. All the used packages are along with the code. In our experiments, we use a single GPU (NVIDIA TITAN Xp) and 8 CPU cores (Intel Xeon E5-2630 v4).

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

# Appendix

# Prefer to Classify:
# Improving Text Classifier via Pair-wise Preference Learning

## A  EXPERIMENTAL DETAILS

### A.1  DATASETS

As described in Section 4.1, we have used the five text classification datasets obtained from the following three different sources, which release the annotation records during the construction of the datasets. Here, all datasets have the annotation records of 5 different annotators for each data; however, the annotators can be different among data, *i.e.*, there are more than 5 annotators in overall. For the tokenization of given data, we commonly set the maximum length $L$ as 256.

**DynaSent** (Potts et al., 2021) is a sentiment classification benchmark with ternary (positive/negative/neutral) sentiments. It is dynamically constructed through multiple iterations of training a classifier model and finding its adversarial samples by involving a human annotator in the loop. In our experiments, we use the dataset from the first round, (1) *DynaSent-R1*, and the dataset from the second round, (2) *DynaSent-R2*. DynaSent-R1 comprises 80,488 training samples, 3,600 validation samples, and 3,600 test samples, respectively. DynaSent-R2 comprises 13,065 training samples, 720 validation samples, and 720 test samples. All the validation and test samples are fully balanced between the three classes. DynaSent dataset and more details of the dataset are officially available at `https://github.com/cgpotts/dynasent`.

**Standford politeness corpus** (Danescu-Niculescu-Mizil et al., 2013) is a binary classification benchmark for predicting whether the given sentence is polite or impolite. Since there are two different input domains within this benchmark, we split them into two different datasets: (3) *Polite-Wiki* from Wikipedia, and (4) *Polite-SE* from Stack Exchange, following the original paper (Danescu-Niculescu-Mizil et al., 2013). Here, two classes: polite and impolite, are defined as the top and, respectively, bottom quartile of sentences when sorted by their politeness score. The classes are therefore balanced, with each class consisting of 1,089 samples for the Wikipedia domain and 1,651 samples for the Stack Exchange domain. We split each dataset into an 8:1:1 ratio to construct training, validation, and test datasets. The source data and more details of the dataset is officially available at `https://www.cs.cornell.edu/~cristian/Politeness.html`.

**Offensive agreement dataset** (Leonardelli et al., 2021) is a binary classification benchmark for predicting whether the given sentence is offensive or not. Each sentence is collected from Twitter using Twitter public APIs, based on the hashtags and keywords on three different domains: Covid-19, US Presidential elections and the Black Lives Matter (BLM) movement. Remarkably, some of the original samples are not available anymore due to the elimination of tweets from the user-side; for example, 10,735 samples are collected initially (Leonardelli et al., 2021), but only 6,513 samples are now available. To address the issue of the reduced number of samples, we slightly modify the dataset to keep the setups of the original paper, *e.g.*, balanced among the classes and domains. Specifically, we gather the given splits of the dataset into the unified one and then re-split it as much be balanced as possible. This re-constructed dataset has 2,400 training samples, 400 validation samples, and 400 test samples. Also, the ratio between Covid-19, Election, and BLM is 3:3:2. The dataset is officially available with the request to authors at `https://github.com/dhfbk/annotators-agreement-dataset`.

**Multi-Genre Natural Language Inference (MultiNLI)** (Sener & Koltun, 2018) is a crowd-sourced collection of 433k sentence pairs annotated with textual entailment information: for a given premise sentence, one should classify whether the given hypothesis sentence is *entailment*, *neutral*, or *contradiction* to the premise (ternary classification). Since the annotation records are only available with validation set, we construct the datasets by splitting it into 8:1:1 for training, validation, and test sets. This re-constructed datasets has 15,717 training samples, 1,964 validation samples, and 1,966 test samples. The source data and more details of the dataset is officially available at `https://cims.nyu.edu/~sbowman/multinli`.

## A.2 BASELINES

We first introduce some notations for a clear explanation. For each sample $\mathbf{x}$, there are annotation records $n_y(\mathbf{x}) \in \mathbb{N}^K$ where $K$ is the number of class and $n_{\texttt{vote}}(\mathbf{x}) = \sum_y n_y(\mathbf{x})$ is the number of votes[6]. Then, the majority voted target label is obtained by finding the most agreed labels, *i.e.*, $y_{\texttt{task}}(\mathbf{x}) = \arg\max_y n_{\texttt{vote}}(\mathbf{x})$, and simply denoted by $y_{\texttt{task}}$. Here, our goal is to train a classifier $f_\theta := W_{\texttt{task}} \circ g_\phi$, composed with Transformer-based language model backbone $g_\phi$ (*e.g.,* BERT (Devlin et al., 2019)) and a random initialized classification head $W_{\texttt{task}}$, where the prediction for $\mathbf{x}$ is obtained with softmax, *i.e.*, $p(\mathbf{x}) = \texttt{Softmax}\big(f_\theta(\mathbf{x})\big)$. For the analysis in Figure 4, we only include four baselines with high performance based on the results in Table 2.

**(1) Vanilla**: as described in Section 2.1, the model $f_\theta$ is trained with the following training loss:

$$\mathcal{L}_{\texttt{train}} = \ell_{\texttt{xe}}(p(\mathbf{x}), y_{\texttt{task}})$$

**(2) Soft-labeling** (Fornaciari et al., 2021): instead of using majority voted label $y_{\texttt{task}}$, it use the soft-labels $q(\mathbf{x}) = n_y(\mathbf{x})/n_{\texttt{vote}}(\mathbf{x})$ with a cross entropy loss:

$$\mathcal{L}_{\texttt{train}} = \ell_{\texttt{xe}}(p(\mathbf{x}), q(\mathbf{x})) = \sum_y -q_y(\mathbf{x}) \log p_y(\mathbf{x})$$

**(3) Margin Loss** (Sharmanska et al., 2016): instead of using majority voted label and cross entropy loss, it use the soft-labels $q(\mathbf{x})$ as a margin for the multi-class hinge loss:

$$\mathcal{L}_{\texttt{train}} = \sum_y \max\{0, q_y(\mathbf{x}) - p_y(\mathbf{x})\}$$

**(4) Filtering** (Leonardelli et al., 2021): following the setups in the original paper, we exclude the ambiguous samples that have a low argeement between the annotators. Specifically, we exclude the samples with $n_{y_{\texttt{task}}} = 3$ since there are 5 annotators for all considered datasets.

$$\mathcal{L}_{\texttt{train}} = \mathbb{1}[n_{y_{\texttt{task}}}(\boldsymbol{x}) > 3]\, \ell_{\texttt{xe}}(p(\mathbf{x}), y_{\texttt{task}})$$

**(5) Weighting** (Uma et al., 2021): using weighted cross entropy that down-weigh the samples with a low argeement:

$$\mathcal{L}_{\texttt{train}} = \mathbf{w}(\mathbf{x})\, \ell_{\texttt{xe}}(p(\mathbf{x}), y_{\texttt{task}})$$

where $\mathbf{w}(\mathbf{x}) = n_{y_{\texttt{task}}}(\mathbf{x})/n_{\texttt{vote}}(\mathbf{x})$.

**(6) Multi-annotator** (Davani et al., 2022): instead of aggregating the different annotators' annotation records, it introduce multiple classification heads $W_{\texttt{task}}^{(t)}$ for learning from each annotator's annotation $y_{\texttt{task}}^{(t)}$. Since each annotator does not annotate all the samples, we simply separate the $n_{\texttt{vote}}(\mathbf{x})$ annotations and train each of classification head where $t = 1, \ldots, n_{\texttt{vote}}$. For the inference of test samples, the ensemble of multiple classification heads is used.

$$\mathcal{L}_{\texttt{train}} = \frac{1}{n_{\texttt{vote}}(\mathbf{x})} \sum_t \ell_{\texttt{xe}}(p^{(t)}(\mathbf{x}), y_{\texttt{task}}^{(t)})$$

where $p^{(t)}(\mathbf{x}) = W_{\texttt{task}}^{(t)} \circ g_\phi(\mathbf{x})$.

**(7) CS-KD** (Yun et al., 2020): for each sample $\mathbf{x}$, the sample $\widehat{\mathbf{x}}$ within the same class, defined by majority voted label $y_{\texttt{task}}$, is also sample and the consistency regularization is additionally imposed between their prediction with a temperature $\tau$. Following the original paper, we use $\tau = 4$.

$$\mathcal{L}_{\texttt{train}} = \ell_{\texttt{xe}}(p(\mathbf{x}), y_{\texttt{task}}) + \ell_{\texttt{xe}}(\widetilde{p}(\mathbf{x}), \widetilde{p}(\widehat{\mathbf{x}}))$$

---

[6]All the used datasets commonly have $n_{\texttt{vote}} = 5$

---

**Algorithm 1** `Prefer-to-Classify (P2C) with extractive preference labels`

---

**Input:** Classifier from a pre-trained language model $f_\theta$, training dataset $\mathcal{D}$ with preference labels $\{(\mathbf{x}^0, \mathbf{x}^1, y_{\texttt{task}}, y_{\texttt{pref}}) | \mathbf{x}^0, \mathbf{x}^1 \in \mathcal{D}\}$, preference predictors $\{h_{\psi^{(t)}}\}_{t=1}^{T}$, mini-batch size $B$, and hyper-parameter $\lambda_{\texttt{cons}}$

---

1: **for** each iteration **do**
2:   Draw a mini-batch $\mathcal{B} = \{(\mathbf{x}_i, y_{\texttt{task},i})_{i=1}^{B}\}$ and the corresponding pairs with preference labels $\widetilde{\mathcal{B}} = \{(\widetilde{\mathbf{x}}_i, y_{\texttt{pref},i})_{i=1}^{B}\}$ from $\mathcal{D}$ with the inconsistency-based sampling (see Section 2.2)
3:   Obtain $f_\theta(\mathbf{x})$ by forwarding $\mathcal{B}$, then calculate $\mathcal{L}_{\texttt{multi}}$ in Eq. 4
4:   Obtain $h_\psi(\mathbf{x})$ by forwarding $\mathcal{B}$ and $\widetilde{\mathcal{B}}$, then calculate $\mathcal{L}_{\texttt{cons}}$ in Eq. 6
5:   Update parameters $\theta$ and $\psi^{(t)}$ to minimize $\mathcal{L}_{\texttt{train}} = \mathcal{L}_{\texttt{multi}} + \lambda_{\texttt{cons}} \mathcal{L}_{\texttt{cons}}$
6: **end for**

---

**Algorithm 2** `Prefer-to-Classify (P2C) with subjective preference labels`

---

**Input:** Classifier from a pre-trained language model $f_\theta$, original training dataset $\mathcal{D} = \{(\mathbf{x}_i, y_i)\}$, collected dataset $\widetilde{\mathcal{D}}$ with preference labels $\{(\mathbf{x}^0, \mathbf{x}^1, y_{\texttt{task}}, y_{\texttt{pref}}) | \mathbf{x}^0, \mathbf{x}^1 \in \mathcal{D}\}$ where $|\widetilde{\mathcal{D}}| = N_{\texttt{pref}}$, preference predictors $\{h_{\psi^{(t)}}\}_{t=1}^{T}$, a mini-batch size $B$ and hyper-parameter $\lambda_{\texttt{cons}}$

---

1: **for** each iteration **do**
2:   Draw a mini-batch $\mathcal{B} = \{(\mathbf{x}_i, y_{\texttt{task},i})_{i=1}^{B}\}$ from $\mathcal{D}$
3:   Draw an another mini-batch $\widetilde{\mathcal{B}} = \{(\mathbf{x}_i, \widetilde{\mathbf{x}}_i, y_{\texttt{task},i}, y_{\texttt{pref},i})_{i=1}^{B}\}$ from $\widetilde{\mathcal{D}}$
4:   Obtain $f_\theta(\mathbf{x})$ by forwarding $\mathcal{B}$, then calculate $\mathcal{L}_{\texttt{multi}}$ in Eq. 4
5:   Obtain $h_\psi(\mathbf{x})$ by forwarding $\widetilde{\mathcal{B}}$, then calculate $\mathcal{L}_{\texttt{cons}}$ in Eq. 5
6:   Update parameters $\theta$ and $\psi^{(t)}$ to minimize $\mathcal{L}_{\texttt{train}} = \mathcal{L}_{\texttt{multi}} + \lambda_{\texttt{cons}} \mathcal{L}_{\texttt{cons}}$
7: **end for**

---

where $\widetilde{p}(\mathbf{x}) = \texttt{Softmax}(f_\theta(\mathbf{x})/\tau)$.

## A.3 PREFER-TO-CLASSIFY (P2C)

In this section, we describe the details of P2C. We first note that the details are slightly different between extractive preference learning (Section 4.1) and subjective preference learning (Section 4.2) due to the difference in experimental setups between them. As described in Section 4, we commonly use $T = 3$ preference heads $\{W_{\texttt{pref}}^{(i)}\}_{i=1}^{T}$ and 2-layer MLPs with `tanh` activation for each $W_{\texttt{pref}}$. We choose hyper-parameters from a fixed set of candidates based on the validation set; $\lambda_{\texttt{pref}} \in \{1.0, 0.1\}$. Also, we only sample the pair of instances with the same majority voted labels for the efficiency.

In case of learning with extractive preference in Section 4.1, we apply the consistency regularization with margin (Eq. 6) by using the difference of annotation as the margin $m$. Specifically, we set a margin of class $y$ between two samples $\mathbf{x}^1$ and $\mathbf{x}^0$ as the difference of their soft-labels $m_y = q_y(\mathbf{x}^1) - q_y(\mathbf{x}^0)$, defined in Section A.2. Then, we apply the consistency regularization to all classes $y \in [0, 1]^K$. In addition, we apply the inconsistency-based sampling for the experiments with extractive preference labels based on the superior experimental results, presented in Section C.

In the case of learning with subjective preference in Section 4.2, we apply the consistency regularization without margin (Eq. 5) since the explicit degree of preference is not given. Also, since the number of pairs with subjective preference labels is limited, we use all of them in training without applying sampling methods described in Section 2.2. We introduce the additional mini-batch from these pairs to optimize the model with consistency regularization. The full procedures of P2C with extractive and subjective preference are described in Algorithm 1 and 2, respectively.

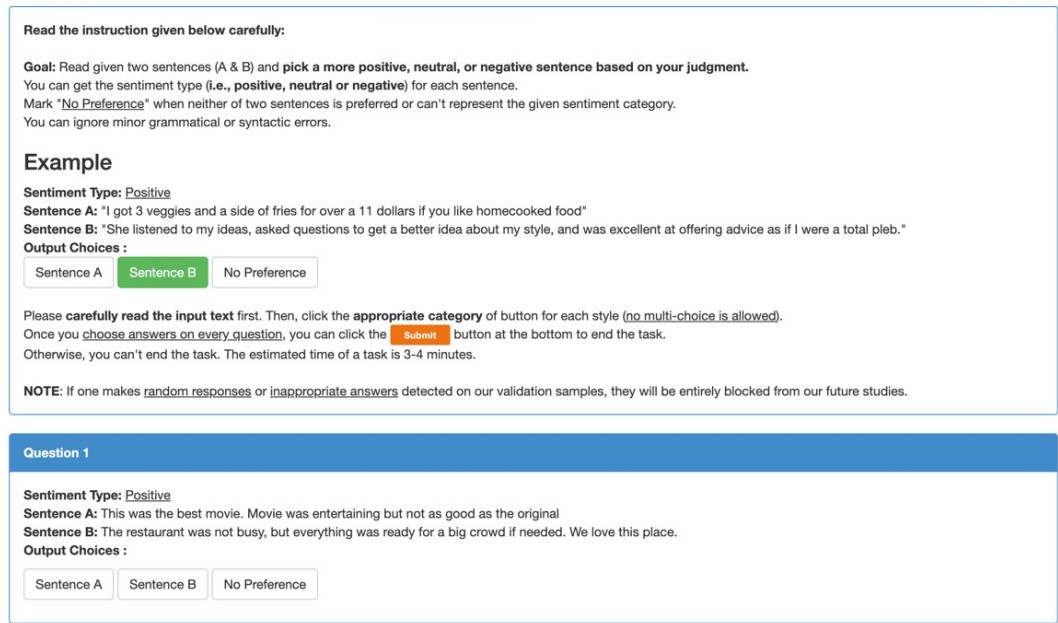

Figure 6: Interface to collect subjective preference labels from crowd workers for sentiment classification based on DynaSent-R2 (Potts et al., 2021).

## B  COLLECTION OF PREFERENCE LABELS

### B.1  INTERFACE

As described in Section 3.2, we gather the subjective preference of the pairs by asking crowd workers to answer "*which sentence is more positive (neutral, or negative)?*" using Amazon's Mechanical Turk crowd-sourcing platform (Crowston, 2012). Figure 6 shows the interface used to collect subjective preference labels from crowd workers for sentiment analysis based on DynaSent-R2 (Potts et al., 2021). The top provides the instructions, and then one example is shown. The whole task has 10 items per Human Interface Task (HIT). Workers were paid US$0.8 per HIT on average, and all workers were paid for their work. To improve the quality of collected preference labels, we only hire the Master workers identified as high-performing workers from Amazon's Mechanical Turk system.

### B.2  MORE DETAILS AND ANALYSIS OF EXTRACTIVE AND SUBJECTIVE PREFERENCE SETS

**Extractive preference.** For a formal description of the process of collecting extractive preference, we borrow some notations introduced in Section A.2. As described in Section 3.1, we obtain the extractive preference label $y_{\texttt{pref}}$ by comparing the number of votes $n_{y_{\texttt{task}}}(\mathbf{x})$ with the given task label $y_{\texttt{task}}$: if $n_{y_{\texttt{task}}}(\mathbf{x}^1) > n_{y_{\texttt{task}}}(\mathbf{x}^0)$, then we assign $y_{\texttt{pref}} = 1$ where it indicates $\mathbf{x}^1 \succ \mathbf{x}^0$. Similarly, we assign $y_{\texttt{pref}} = 0$ when $n_{y_{\texttt{task}}}(\mathbf{x}^1) < n_{y_{\texttt{task}}}(\mathbf{x}^0)$ and $y_{\texttt{pref}} = 0.5$ when $n_{y_{\texttt{task}}}(\mathbf{x}^1) = n_{y_{\texttt{task}}}(\mathbf{x}^0)$, respectively. To reduce the noisy signal and focus on the effective pair, we only compare the samples that have the same majority voted labels, *i.e.*, $y_{\texttt{task}}(\mathbf{x}^1) = y_{\texttt{task}}(\mathbf{x}^0)$. The resulting distribution of extractive preference labels for each data is presented in Figure 7 and 8.

**Subjective preference.** To address the limitation of the extractive preference from the annotation records, we collect the subjective preference labels from crowd workers using the inference in Figure 6. As described in Section 3.2, we dynamically collect the subjective preference labels. Here, to increase the effectiveness of the constructed preference dataset, we query the pairs with the unique anchor sample where $\mathbf{x}^1$ is denoted as an anchor sample of given pair $(\mathbf{x}^1, \mathbf{x}^0)$. Also, we only query the pairs of the samples that have the same majority voted labels as in the case of extractive preference labels. Figure 5(a) shows the distribution of collected preference labels. We remark that the collected

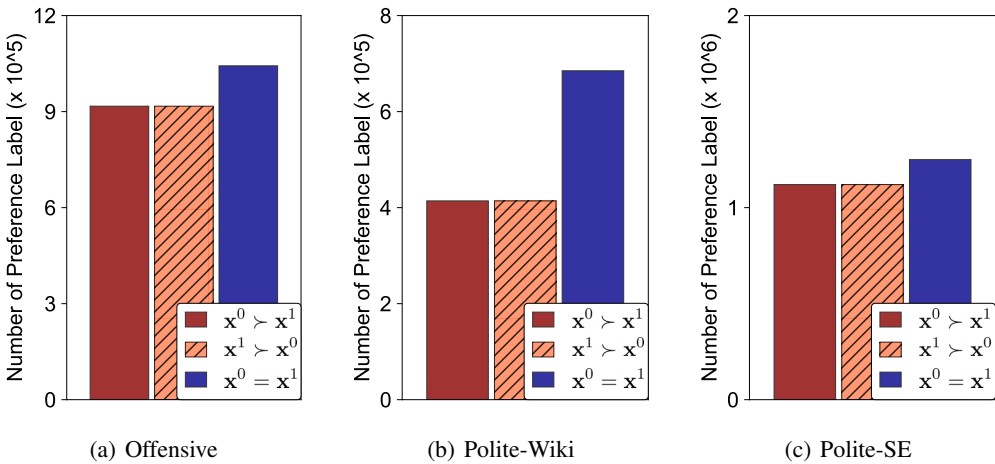

Figure 7: Distribution of the extractive preference labels from the annotation records.

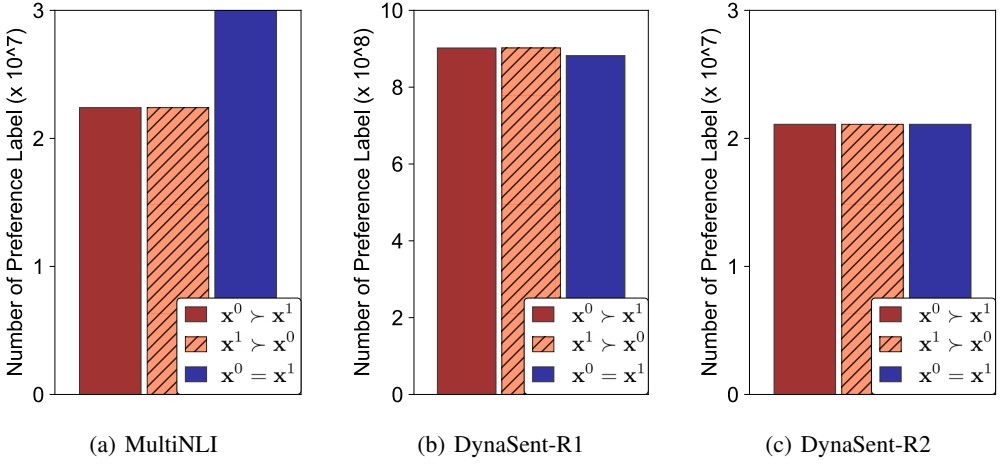

Figure 8: Distribution of the extractive preference labels from the annotation records.

subjective preference labels are different compared to the extractive ones: collected subjective preferences shows 40.3 % correspondence with the extractive preference labels, respectively.

### B.3 MORE EXAMPLES IN EXTRACTIVE AND SUBJECTIVE PREFERENCE SETS

In Table 5, we present more examples in our extractive and subjective preference sets on DynaSent-R2 dataset, similar to Table 1.

Table 5: More examples in our extractive and subjective preference sets on DynaSent-R2.

| | |
|---|---|
| **A**: The prices were great and the service was friendly, which is why I'm surprised by my overall feeling of the place. | **B**: I entered the college apartment and was shocked, everything was neat and tidy. |
| Sentiment: Positive, Extractive Preference: **B** ≻ **A**, Subjective Preference: **A** ≻ **B** | |
| **A**: The atmosphere is great, Nikko our server was personable, knowledgeable and just all-around great, and the food is looking so delicious. | **B**: The loaded fries were to die for. |
| Sentiment: Positive, Extractive Preference: **No preference**, Subjective Preference: **A** ≻ **B** | |
| **A**: The food was surprisingly similar to a chain restaurant such as applebee's or Friday's. | **B**: The soup and pad Thai arrived soon after we ordered. |
| Sentiment: Neutral, Extractive Preference: **A** ≻ **B**, Subjective Preference: **B** ≻ **A** | |
| **A**: I have a laptop but it acts like a phone. | **B**: The baby cooed and laughed without understanding the gravity of the situation. |
| Sentiment: Neutral, Extractive Preference: **A** ≻ **B**, Subjective Preference: **B** ≻ **A** | |
| **A**: Went on a Thursday night with my two boys, got a table right away close to the kitchen. | **B**: They provided great service and did a wonderful job with my greasy hair and heavy makeup. |
| Sentiment: Positive, Extractive Preference: **No Preference**, Subjective Preference: **B** ≻ **A** | |
| **A**: We requested a new tech and that guy said it was a major leak and the would need to charge $7,000. | **B**: Teacher warned the teenagers not to enter the short cut because trespassers will be severely punished |
| Sentiment: Neutral, Extractive Preference: **No Preference**, Subjective Preference: **A** ≻ **B** | |
| **A**: The food is not fresh and delicious. | **B**: They treat customers like they want their money. |
| Sentiment: Negative, Extractive Preference: **A** ≻ **B**, Subjective Preference: **B** ≻ **A** | |
| **A**: After check in I had to search for the room without any direction. It was horrible. | **B**: That food blender worked great, as a paper weight. |
| Sentiment: Negative, Extractive Preference: **No Preference**, Subjective Preference: **A** ≻ **B** | |
| **A**: The menu choices were too narrow. | **B**: The waiter seem to be busy else where in the back and would show up a few moments after I was started to wonder if he'd left. It was awful. |
| Sentiment: Negative, Extractive Preference: **No Preference**, Subjective Preference: **B** ≻ **A** | |

### B.4 EXAMPLES OF EXTRACTIVE PREFERENCE SETS ON DIFFERENT DATASETS

In this section, we present more examples of extractive preference sets on another datasets used in Section 4.1. Specifically, we present the examples of Polite-SE dataset in Table 6 and the examples of MultiNLI in Table 7, respectively.

Table 6: Examples of the collected extractive preference sets on Polite-SE dataset.

| | |
|---|---|
| **A**: @Praetorian: You had me into tears. I agree to you but is that going to help me with my question? | **B**: This is not really enough to tell what can be the problem. Have you tried debug you code, for example trace something in the conditions (so the function what have to be called is called properly or not)? |

Politeness: Impolite, Extractive Preference: **A ≻ B**

| | |
|---|---|
| **A**: Usually compilers should generate a good code for such algorithms. Did you check what assembly a C compiler is generating? | **B**: Why do you want to view the code? Maybe if you give a better description of what you want to do? :) |

Politeness: Polite, Extractive Preference: **No Preference**

| | |
|---|---|
| **A**: Please don't downvote without stating your reason. Should this perhaps be turned into a wiki question? | **B**: Homework? Who told you that checking the pre-condition is a bad idea? |

Politeness: Impolite, Extractive Preference: **B ≻ A**

| | |
|---|---|
| **A**: He has said its for ASP! or am I missing something? | **B**: Sorry but I never saw any application breaking after deleting/moving.folders... usually they just reconfigure and restore to defaults. What exactly did break for you? |

Politeness: Impolite, Extractive Preference: **No Preference**

| | |
|---|---|
| **A**: Could you describe the problems you are having? Are you logging the response codes? | **B**: @den: OK, so the rectangles are represented as two points. Is this a vector representation or a raster? |

Politeness: Polite, Extractive Preference: **A ≻ B**

Table 7: Examples of the collected extractive preference sets on MultiNLI dataset.

| | |
|---|---|
| **A**: (premise) This is arguably starting to distort the practice of science itself. (hypothesis) This began to distort scientific practice. | **B**: (premise) This number represents the most reliable, albeit conservative, estimate of cases closed in 1999 by LSC grantees. (hypothesis) This is an actual verified number of closed cases. |

Textual Entailment: Entailment, Extractive Preference: **A ≻ B**

| | |
|---|---|
| **A**: (premise) Uh-huh well maybe well i've enjoyed talking to you okay bye-bye. (hypothesis) I liked talking to you. | **B**: (premise) And then I was off, the world exploding behind me. (hypothesis) The world exploded behind me. |

Textual Entailment: Entailment, Extractive Preference: **No Preference**

| | |
|---|---|
| **A**: (premise) I am surprised though that we do have so many that are in politics down here (hypothesis) I am surprised that not many of them are in politics down here. | **B**: (premise) Outside, set in manicured gardens, are the remains of the Abbey of Holyrood. (hypothesis) The gardens containing the remains of the Abbey of Holyrood are in disarray and not well-kept. |

Textual Entailment: Contradiction, Extractive Preference: **No Preference**

| | |
|---|---|
| **A**: (premise) After the purge of foreigners, only a few stayed on, strictly confined to Dejima Island in Nagasaki Bay. (hypothesis) A few foreigners were left free after the purge on foreigners. | **B**: (premise) It sure will well good to talk to. (hypothesis) That is unlikely and this conversation has gotten us nowhere. |

Textual Entailment: Contradiction, Extractive Preference: **B ≻ A**

| | |
|---|---|
| **A**: (premise) Trying Your Luck. (hypothesis) Think carefully and calculate your way to a certain victory. | **B**: (premise) Emeralds? (hypothesis) Are they wearing emeralds? |

Textual Entailment: Neutral, Extractive Preference: **B ≻ A**

# C    MORE ABLATION STUDY

In this section, we provide more ablation studies on the design choices of P2C. Here, all experiments are conducted on DynaSent-R2 (Potts et al., 2021) and Offensive (Leonardelli et al., 2021) datasets with extractive preference labels, as same as we have done in Section 4.1. The values and error bars are the mean and standard deviation across five random seeds. The results with the chosen design in Section 4.1 are underlined.

**Multiple Preference head for preference learning.** In Section 2.2, we introduce multi-preference heads with diversity regularization (Eq. 3) to effectively learn the given preference labels. To see the effect, we compare it with two different designs for preference heads: 1) single-preference head and 2) multi-preference heads without diversity regularization. Remark that the other components, consistency regularization, and inconsistency-based sampling, are still applied to separately verify the effect from different designs of the preference head. As shown in Table 8, one can verify that a single preference head is not enough to exploit the given preference labels fully; hence, the empirical gain is relatively small compared to multi-preference heads. Also, it is observable that the proposed regularization is more effective to impose the diversity than only relying on the random initialization.

Table 8: Effect of different designs for preference head.

| Dataset | Single-Pref Head | Multi-Pref Heads without diversity | Multi-Pref Heads with diversity |
|---|---|---|---|
| DynaSent-R2 | $72.22_{\pm0.55}$ | $72.75_{\pm0.42}$ | $\underline{73.06_{\pm0.31}}$ |
| Offensive | $77.08_{\pm0.57}$ | $77.25_{\pm0.92}$ | $\underline{77.81_{\pm0.21}}$ |

**Auxiliary loss for preference learning.** As described in Section 2.2, we use a consistency regularization (Eq. 5 and 6) between classification and preference learning as an auxiliary loss for learning preference; specifically, consistency regularization with margin (Eq. 6) is used in Section 4.1. To clarify the effectiveness of this regularization, we compare it with 1) consistency regularization without margin (Eq. 5). We also compare it 2) soft-labeling, which also uses the annotation records to construct soft-labels instead of the preference and margin. Here, we use random sampling instead of inconsistency-based sampling since it is designed explicitly for consistency regularization while using the multi-preference heads. Table 9 shows the results of these auxiliary losses; although consistency regularization is effective in improving the performance without margin, the gain is smaller than the consistency regularization with margin since the latter utilizes the additional knowledge about the given preference label. In addition, the result with soft-labeling validate that the gain from our consistency loss is not from the use of the annotation records but from the regularization that imposes the following intuition: *more preferred instance should have a higher confidence from the classifier*.

Table 9: Effect of different auxiliary losses to learn from the given preference labels.

| Dataset | Soft -labeling | Consistency without margin | Consistency with margin |
|---|---|---|---|
| DynaSent-R2 | $72.29_{\pm0.88}$ | $72.40_{\pm0.71}$ | $\underline{72.67_{\pm0.89}}$ |
| Offensive | $77.04_{\pm1.05}$ | $77.54_{\pm0.95}$ | $\underline{77.67_{\pm0.99}}$ |

**Sampling of pairs for preference learning.** To improve the efficiency of preference learning by sampling the informative pairs during the training, we introduce two advanced sampling methods: (1) *disagreement-based* sampling and (2) *inconsistency-based sampling* in Section 2.2. Remark that the other components, consistency regularization with margin and multi-preference heads, are still applied to verify the effect from different sampling methods separately. In Table 10, we compare both sampling methods to random sampling. Here, one can verify that both ways are more effective than random sampling, and the inconsistency-based sampling is slightly better than the disagreement-based sampling. Hence, we commonly used inconsistency-based sampling for in Section 4.1.

Table 10: Effect of different sampling methods with preference learning.

| Dataset | Random | Disagreement | Inconsistency |
|---|---|---|---|
| DynaSent-R2 | $72.67_{\pm 0.89}$ | $72.73_{\pm 0.66}$ | $\underline{73.06}_{\pm 0.31}$ |
| Offensive | $77.67_{\pm 0.99}$ | $77.75_{\pm 1.49}$ | $\underline{77.81}_{\pm 0.21}$ |

**Sensitivity to $\mathcal{L}_{\text{div}}$.** To verify the sensitivity of our method with $\mathcal{L}_{\text{div}}$, we conduct the experiments by introducing $\lambda_{\text{div}}$, a coefficient of $\mathcal{L}_{\text{div}}$, and varying it to investigate its effect. In Table 11, one can observe that KL divergence does not dominate the entire loss until the certain level of $\lambda_{\text{div}}$ including the original value ($\lambda_{\text{div}}=1$), but it can diverge with too large value (*e.g.*, $\lambda_{\text{div}} = 10$). Hence, we recommend to use the original value or investigate $\lambda_{\text{div}}$ with smaller than 1.

Table 11: Effect of diversity regularization between multi-preference heads with $\lambda_{\text{div}}$.

| Dataset | $\lambda_{\text{div}} = 0$ | $\lambda_{\text{div}} = 1$ | $\lambda_{\text{div}} = 2$ | $\lambda_{\text{div}} = 10$ |
|---|---|---|---|---|
| DynaSent-R2 | $72.75_{\pm 0.42}$ | $\underline{73.06}_{\pm 0.31}$ | $71.44_{\pm 0.68}$ | $57.05_{\pm 2.14}$ |
| Offensive | $77.25_{\pm 0.92}$ | $\underline{77.81}_{\pm 0.21}$ | $75.35_{\pm 1.03}$ | $65.05_{\pm 6.70}$ |

# D  ADDITIONAL EXPERIMENTAL RESULTS

**Smaller training samples.** Here, we validate the effectiveness of P2C with extractive preferences for the smaller training samples. Specifically, we control the number of training samples ($N$) of the DynaSent-R2 dataset from $N = 250$ to $N = 4000$, and compare our method with three representative baselines with high performance: Vanilla, Soft-labeling, and Multi-annotator. As shown in Table 12, P2C shows significant improvement, especially when the dataset size is smaller. We also remark that P2C shows consistent improvement for all cases while other baselines do not.

Table 12: Results with the smaller training samples.

| Method | $N = 250$ | $N = 500$ | $N = 1000$ | $N = 2000$ | $N = 4000$ |
|---|---|---|---|---|---|
| Vanilla | $54.89_{\pm2.46}$ | $60.36_{\pm2.98}$ | $63.61_{\pm0.92}$ | $66.50_{\pm0.76}$ | $68.69_{\pm1.41}$ |
| Soft-labeling | $57.75_{\pm2.35}$ | $60.03_{\pm1.46}$ | $62.81_{\pm1.45}$ | $66.78_{\pm1.16}$ | $68.17_{\pm1.09}$ |
| Multi-annotator | $57.33_{\pm3.23}$ | $61.39_{\pm1.76}$ | $63.00_{\pm0.87}$ | $66.19_{\pm0.84}$ | $68.78_{\pm1.46}$ |
| P2C (Ours) | $\mathbf{58.94}_{\pm1.16}$ | $\mathbf{61.83}_{\pm1.15}$ | $\mathbf{64.13}_{\pm1.04}$ | $\mathbf{67.72}_{\pm0.46}$ | $\mathbf{69.83}_{\pm0.64}$ |

**Semi-supervised learning with the collected preference labels.** As the collection of preference labels induce the additional cost, especially with subjective preference, it would be beneficial if there is another way to fully exploit the collected labels; hence, we validate that the gain from limited preference labels can be further enlarged with the semi-supervised learning scenario. We remark that semi-supervised learning is well known to be effective for better utilization of limited labeled data (Berthelot et al., 2019; Kim et al., 2020). To verify the effectiveness of this approach, we conduct the following experiments: using trained P2C models with given 5,000 preference labels, we generate pseudo labels for the preference of each pair. Here, we clarify that we generate "pseudo" preference labels (i.e., pair-wise) for unlabeled pair data for semi-supervised learning while we assume that task labels (i.e., instance-wise) for all training data are still available. Then, we train another model with P2C using these "pseudo preference labels". As summarized in Table 13, we observe the clear improvements with this approach, and the gain is much larger with subjective preference, which indicates the informativeness of these preference labels. We also remark that there is still room for improvement from the advanced semi-supervised learning approaches such as iterative refinement (Ziegler et al., 2019) or confidence thresholding (Park et al., 2022).

Table 13: Semi-supervised learning with the collected preference labels with pseudo-labeling method.

| Method | Vanilla | Extractive Preference | Subjective Preference |
|---|---|---|---|
| Original | $71.23_{\pm1.05}$ | $71.94_{\pm1.96}$ | $72.40_{\pm0.62}$ |
| Semi-sup | $71.23_{\pm1.95}$ | $72.11_{\pm0.92}$ | $\mathbf{73.08}_{\pm0.42}$ |

**Compatibility with other types of models.** While we have previously used a model built over RoBERTa-base (Liu et al., 2019), the proposed P2C is not limited to the specific model. To verify this, we conduct additional experiments based on DynaSent-R2 with extractive preference labels from the annotation records. As shown in Table 14, the proposed P2C consistently improves the test accuracy of classifiers built over one-hot encoded vectors (Galke & Scherp, 2022) as well as other language models (BERT-base (Devlin et al., 2019), ALBERT-base (Lan et al., 2020), and RoBERTa-large).

Table 14: Results with other types of models.

| Method | One-hot encoding | BERT-base | ALBERT-base | RoBERTa-large |
|---|---|---|---|---|
| Vanilla | $54.42_{\pm0.77}$ | $67.26_{\pm1.15}$ | $62.72_{\pm0.73}$ | $75.62_{\pm0.60}$ |
| P2C (Ours) | $55.47_{\pm0.82}$ | $68.26_{\pm0.56}$ | $65.00_{\pm1.13}$ | $77.71_{\pm0.36}$ |

**Different number of annotations.** In addition, we validate that pair-wise human preference could be a more efficient annotation than the existing instance-wise task labels. To this end, we conduct

additional experiments on DynaSent-R2 with subjective preference labels by varying the number of additional annotations ($N$) upon the existing 7.5k task labels, similar to Table 4.

As shown in Table 15, it is verified that the pair-wise human preference even requires fewer annotations to achieve the same level of test accuracy; for example, it requires 29.8% fewer annotations than instance-wise task label to achieve 70.5% test accuracy. Here, we estimate the number of annotations at 70.5% test accuracy by interpolating the results with polynomial approximation (Pedregosa et al., 2011).[7] Also, we note the proposed framework has an additional advantage; it could provide an effective solution, especially when input data acquisition is expensive, such as medical datasets or human-in-the-loop benchmarks.

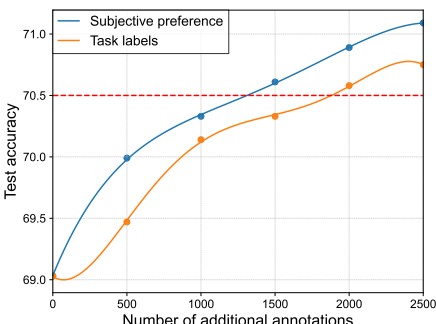

Figure 9: Interpolated results of Table 15

Table 15: Results with different number of additional annotations ($N$).

| Method | $N = 0$ | $N = 500$ | $N = 1000$ | $N = 1500$ | $N = 2000$ | $N = 2500$ |
|---|---|---|---|---|---|---|
| Task Labels | $69.03_{\pm 1.29}$ | $69.47_{\pm 0.58}$ | $70.14_{\pm 0.50}$ | $70.33_{\pm 0.90}$ | $70.58_{\pm 0.73}$ | $70.75_{\pm 1.29}$ |
| Subjective Preference | $69.03_{\pm 1.29}$ | $69.99_{\pm 1.32}$ | $70.33_{\pm 0.85}$ | $70.61_{\pm 1.31}$ | $70.89_{\pm 0.75}$ | $71.09_{\pm 1.55}$ |

---

[7]We use 5 degree for interpolation, and code is available at https://scikit-learn.org/stable/auto_examples/linear_model/plot_polynomial_interpolation.html

