# OpenReview forum: "Prefer to Classify: Improving Text Classifier via Pair-wise Preference Learning"
_ICLR.cc/2023/Conference — Submitted to ICLR 2023_

### Official Review · Reviewer_cBxH · 2022-10-24

**Confidence:** 3
**Correctness:** 3
**Technical Novelty And Significance:** 2
**Empirical Novelty And Significance:** Not applicable
**Recommendation:** 5

**Clarity, Quality, Novelty And Reproducibility:**

In terms of clarity, the logic of this article is well organized, but some expressions need to be corrected.
In terms of novelty, P2C is not the first method to utilize the annotators’ disagreement from the annotation records.
In terms of reproducibility, since some of the labels required for model training (i.e., labels for preference learning task) need to be manually labeled, it requires huge human resources to reproduce this work from zero. Some of the experimental results provided by the author cannot be reproduced if the data provided by the author (if any) is used directly instead of labeling the data, such as in Fig. 5(b).
In terms of quality, this paper proposes a novel MTL framework, however, in some cases (i.e., datasets without any remaining annotation records), it needs to take a huge human resource to achieve performance improvement.

**Strength And Weaknesses:**

Strengths
1. The authors formulate a preference learning task as a classification task and propose a framework to simultaneously solve text classification tasks and preference learning tasks. These formulations are clean and make the difference between the various setup clear.
2. Improved performance (with test accuracy) compared to baseline methods.
3. The author has done a lot of experiments to verify the importance of each module in the P2C framework.

Weaknesses:
1. The description in parts of the paper needs to be improved.
- The abstract part mentions "we propose a new multi-task learning framework", the introduction part mentions "we propose a new multi-task learning method", and the conclusion part mentions "we propose a novel multi-task learning object". It is not a reasonable way to introduce P2C in three ways.
- The information expressed in Eq. 5 is that the preference learning task is a binary classification task, but in the authors’ setting, the preference learning task is a triple classification task. That is, Eq. 5 does not express an equally preferable case (i.e., = 0.5).
2. For datasets without any remaining annotation records, the proposed framework requires huge human resources (labels for preference learning tasks) to construct labels for preference learning tasks. Is this measure of improving model performance at a huge human resource desirable?

**Summary Of The Paper:**

In the paper, the authors propose P2C, a framework to synthesize the classification and preference learning tasks. The authors argue that capturing the preference information from the annotators during the data labeling process would lead to better text classification task performance. They propose two methods to collect annotator's preference information: (1) Implicit preference extraction on existing annotation records; (2) Explicit preference collection of annotators by online query and they formulate preference learning as a supervised learning problem.

**Summary Of The Review:**

The MTL design to combine main tasks (e.g., text classification task in the paper) and other useful tasks (e.g., preference learning task in the paper) is well motivated. However, reproduction in some cases requires huge human resources.

---

### Official Review · Reviewer_TEZ4 · 2022-10-25

**Confidence:** 3
**Correctness:** 4
**Technical Novelty And Significance:** 2
**Empirical Novelty And Significance:** 3
**Recommendation:** 8

**Clarity, Quality, Novelty And Reproducibility:**

Clarity: Good.

Quality: Good.

Novelty: Good.

Reproducibility: Good.

**Strength And Weaknesses:**

Strength:
  - The idea is simple yet effective.
  - The paper is well-motivated and well-written.
  - The empirical results are convincing.
  - Besides simply using annotation records, the authors also employ crowd workers to provide specific preference annotation. And they show using the preference labels further improves the model performance compared to simply extracting preferences from annotation records, making the logic and motivation sound and self-explained.
  - The paper illustrates the effectiveness of pair-wise human preference to better guide the model, which may motivate better data annotation in the future.

Weakness:
  - Since only a few datasets provide annotation record and collecting human annotation is very expensive, the application scenario of this model is very limited at this time.

**Summary Of The Paper:**

This paper presents a novel framework that utilizes the annotation disagreements in human-annotated benchmarks and human-annotated preference to improve the model training in scenarios where annotation records are provided.

**Summary Of The Review:**

This paper proposes a simple yet effective way to utilize human annotation records and incorporate human preference to guide the model training, which may motivate better data annotation and data usage in the future. I would recommend an acceptance for this paper.

---

### Official Review · Reviewer_M4mz · 2022-11-02

**Confidence:** 4
**Correctness:** 3
**Technical Novelty And Significance:** 2
**Empirical Novelty And Significance:** 2
**Recommendation:** 3

**Clarity, Quality, Novelty And Reproducibility:**

As stated earlier, the paper is easy to follow and the authors have explained the problem setting, their solution and provided experimental results.
In terms of novelty, it is hard to judge whether the framework itself is novel since several preference learning that perform pair-wise comparisons exist.
The authors have provided details about implementing the approach and for using the datasets - which help with reproducibility. However, it is not clear if the questions they asked and the responses they obtained will be identical if another person attempted to reproduce the results.

**Strength And Weaknesses:**

Overall, the paper is well written, and the authors have explained their approach and provided experimental results to show their work.

However, there are few weaknesses.
There are similar approaches in the literature, studied under various titles such as active learning, learning to rank and even preference learning. The approach sounds interesting, but not necessarily novel. The authors have compared their results against 7 approaches, but it is not clear which baseline is the state-of-art approach. Further, referencing approaches from active learning and learning to rank would greatly help demonstrating the effectiveness of the work.

While the authors have given examples for the DynaSent-R1 dataset, it is not clear what kind of questions/ preferences were obtained for other datasets. Since they are subjective in nature, it will be good to understand the nature of responses obtained on others as well. On the positive side, the results are encouraging.

Finally, in terms of relevance to the conference: the paper is more of an active learning or preference learning paper, and there is less focus on representations themselves. Though the authors have used a model build over RoBERTa-base, their approach could probably work on one-hot encoded vectors too.

**Summary Of The Paper:**

The authors present a multi-task learning framework called  prefer-to-classify (P2C), which is based on pair-wise preference learning. Specifically, their work is focused on handling NLP tasks. The paper explains the overall approach, and the process of selecting informative pairs, and how the preferences are collected from human annotators using Amazon Turk.
They have shown experimentally  that their approach is able to outperform on baselines - they have used 7 base lines across 6 datasets.

**Summary Of The Review:**

The authors present a multi-task learning framework called  prefer-to-classify (P2C), for handling NLP tasks. They have shown experimentally  that their approach is able to outperform on baselines - they have used 7 base lines across 6 datasets.
There are some drawbacks in terms of comparison baselines as discussed in the weakness section.

---

### Decision · Program_Chairs · 2023-01-20

**Decision:**

Reject

**Justification For Why Not Higher Score:**

While there were many positive aspects to this paper, there was a general consensus regarding the need for additional contextualization wrt closely related concepts (e.g., learning to rank for text settings), discussion of practical considerations, and expected impact within the broader ICLR community.

**Justification For Why Not Lower Score:**

N/A

**Metareview: Summary, Strengths And Weaknesses:**

The authors propose a multi-task formulation for incorporating 'standard' (class membership) labels and 'preference' (pairwise class membership intensity) labels for text classification tasks -- referred to as "prefer to classify" (P2C). Specifically, in text classification tasks where some examples have stronger membership to a particular class (e.g., in sentiment analysis, some examples may be 'more' positive), the authors propose collecting pairwise preferences via (1) voting information from full annotation records (i.e., how many annotators labeled as positive) and (2) explicit pairwise labeling from annotators. Once this information is collected (from crowd workers), it is incorporated into a multi-task learning formulation that incorporates both classification & preference learning loss, diverse multi-preference heads in the transformer network, and consistency regularization to bias toward preference ordering consistency. They also propose using disagreement-based sampling and inconsistency-based sampling to query for preference information. Experiments are conducted on six text classification/entailment datasets against seven recent, suitable baselines and show consistent performance improvements.

The consensus strengths of this work include:
- The paper is well-written and easy to understand; preference information is feasible to obtain and performs well on these tasks.
- The specific multi-task formulation was thoroughly considered, including multiple variants that are evaluated via an ablation study.
- The primary empirical results are consistently positive and there are sufficient secondary experiments to support the proposed methods and claims stated in the paper (which is further extended in the rebuttal to show embedding-invariance in performance improvements).

The weaknesses of this work from the reviews (and my own reading) included:
- The novelty relative to general learning to rank framework as applied to text-relevance tasks is warranted. Specifically, one interpretation of search is to rank documents wrt their relevance to a particular query (which would analogously be the class label -- in the limit, we would be blurring the line between search and massively multiclass text classification which is generally studied within the search community; e.g., [Chang, et al., Taming Pretrained Transformers for Extreme Multi-label Text Classification, KDD20]). There are distinguishing features specific to each problem, but it isn't orthogonal either. Also, this has been addressed in some works (e.g., [Atapour-Abarghouei, Bonner & McGough, Rank over Class: The Untapped Potential of Ranking in Natural Language Processing, 2021]).
- The relationship of the sampling method to selective sampling (i.e., active learning) isn't negligible as the combinatorics of pairwise vs. pointwise methods is an important consideration. In this line, disagreement and consistency-based querying functions have been proposed (e.g., [Muslea, Minton & Knoblock, Active + Semi-Supervised Learning = Robust Multi-View Learning, ICML02]).
- This work is shown in the context of text classification in cases where there is a 'strength of membership' variance across examples in the same class. While this is a large class of text classification problems (and it is also show to work for entailment), in principle, this work shouldn't be limited to text classification -- but any such setting. Since ICLR is application-agnostic, a stronger paper would consider settings beyond text classification. This isn't really a weakness, but more of a missed opportunity.

Overall, the idea is good, the formulation is sensible, and the demonstrated empirical performance is solid (and thorough). The primary concerns were regarding better contextualization relative to related work for learning to rank in other text-based settings, discussion regarding practical concerns of collecting pairwise preferences (which is studied in preference elicitation work), and potentially increasing the scope of the work beyond text classification to make for a stronger contribution.